# Modulation of mRNA and lncRNA expression dynamics by the Set2–Rpd3S pathway

Ji Hyun Kim[1,2,*], Bo Bae Lee[1,2,*], Young Mi Oh[1,2], Chenchen Zhu[3,4,5], Lars M. Steinmetz[3,4,5], Yookyeong Lee[1], Wan Kyu Kim[1], Sung Bae Lee[6], Stephen Buratowski[7] & TaeSoo Kim[1,2]

H3K36 methylation by Set2 targets Rpd3S histone deacetylase to transcribed regions of mRNA genes, repressing internal cryptic promoters and slowing elongation. Here we explore the function of this pathway by analysing transcription in yeast undergoing a series of carbon source shifts. Approximately 80 mRNA genes show increased induction upon SET2 deletion. A majority of these promoters have overlapping lncRNA transcription that targets H3K36me3 and deacetylation by Rpd3S to the mRNA promoter. We previously reported a similar mechanism for H3K4me2-mediated repression via recruitment of the Set3C histone deacetylase. Here we show that the distance between an mRNA and overlapping lncRNA promoter determines whether Set2–Rpd3S or Set3C represses. This analysis also reveals many previously unreported cryptic ncRNAs induced by specific carbon sources, showing that cryptic promoters can be environmentally regulated. Therefore, in addition to repression of cryptic transcription and modulation of elongation, H3K36 methylation maintains optimal expression dynamics of many mRNAs and ncRNAs.

[1] Department of Life Science, Ewha Womans University, Seoul 03760, Korea. [2] The Research Center for Cellular Homeostasis, Ewha Womans University, Seoul 03760, Korea. [3] Genome Biology Unit, European Molecular Biology Laboratory, Meyerhofstrasse 1, 69117 Heidelberg, Germany. [4] Stanford Genome Technology Center, Stanford University School of Medicine, Stanford, California 94305, USA. [5] Department of Genetics, Stanford University School of Medicine, Stanford, California 94305, USA. [6] Department of Brain and Cognitive Sciences, DGIST, Daegu 42988, Korea. [7] Department of Biological Chemistry and Molecular Pharmacology, Harvard Medical School, Boston, Massachusetts 02115, USA. * These authors contributed equally to this work. Correspondence and requests for materials should be addressed to S.B. (email: steveb@hms.harvard.edu) or to T.K. (email: tskim@ewha.ac.kr).

Eukaryotic transcription is regulated by post-translational modifications of histones, including acetylation, methylation, phosphorylation and ubiquitination[1,2]. Histone acetylation can promote RNA polymerase II (RNApII) transcription by disrupting the interaction between histone proteins and DNA or by recruiting factors that regulate chromatin structure and transcription. Histone acetylation is a highly dynamic modification regulated by both histone acetyltransferases and histone deacetylases (HDACs). Site-specific histone methylations also play important roles in regulating histone acetylation levels by targeting histone acetyltransferases or HDACs to specific gene locations[3].

The Set2 methyltransferase directly binds elongating RNApII and co-transcriptionally methylates H3K36 (refs 4–7). In early stages of transcription, the basal transcription factor TFIIH phosphorylates serine 5 of the C-terminal domain (CTD) of Rpb1, the largest subunit of RNApII, creating binding sites for factors involved in transcription initiation, histone modification and early termination[8]. During transcription elongation, Ctk1 kinase phosphorylates serine 2 of CTD to create a binding site for the Set2 methyltransferase[4,5,7,9]. In general, H3K36me2 and H3K36me3 are enriched in transcribed regions with increased levels towards 3′-ends, and this pattern is highly conserved in all eukaryotes[10–13]. Methylated histone tails primarily function as binding sites for downstream effector proteins that have chromodomains, tudor domains or Plant Homeodomain (PHD) fingers[1,14].

Nucleosomes methylated on H3K36 are deacetylated by the Rpd3 small complex, Rpd3S (refs 15–17). The Eaf3 chromodomain and the Rco1 PHD finger mediate binding of Rpd3S to histones and this binding is stimulated by H3K36 methylation[18]. Although Rpd3S may be in part recruited to transcribed regions via interaction with the phosphorylated CTD of RNApII, histone deacetylation by this complex requires H3K36 methylation[19,20]. Deacetylation by the Set2–Rpd3S pathway represses transcription from cryptic internal promoters and slows transcription elongation[15,16,21–23]. However, a recent study showed that the Set2–Rpd3S pathway had little effect on mRNA levels under steady-state laboratory growth conditions[24].

Here we describe a new function for the Set2–Rpd3S in modulating mRNA gene expression dynamics during induction. Microarray analysis shows that a majority of Set2-repressed genes are overlapped by long non-coding RNA (lncRNA) transcription, from either an upstream or an antisense promoter. This non-coding transcription targets H3K36me3 and the Rpd3S HDAC to the Set2-repressed promoters. In addition, new Set2–Rpd3S-repressed cryptic promoters are seen that are induced by specific carbon sources. Therefore, Set2–Rpd3S regulates both mRNA promoters and cryptic non-coding RNA (ncRNA) promoters. We previously showed that overlapping lncRNA transcription could also target H3K4me2 and the Set3C HDAC to repress mRNA induction[25]. We find that a Set2-repressed promoter can be switched to Set3 repression by reducing the distance from the lncRNA promoter.

## Results

**Set2 negatively affects gene induction kinetics.** To investigate how H3K36 methylation affects mRNA transcription, gene expression was analysed in SET2 and set2Δ cells during carbon source shifts (Fig. 1a). Cells were pre-cultured in synthetic complete medium containing raffinose (0 time point) and then shifted to galactose (15, 30, 60 and 120 min), glucose (15, 30, 60 and 120 min) and then back to galactose (15 and 30 min). Under these conditions, ∼800 genes were significantly induced or repressed at some point in the time course[25]. While the HPF1

transcript levels remained steady, GAL1 and GAL7 were strongly induced in galactose media and repressed by glucose as expected (Supplementary Fig. 1). Conversely, PYK1 was downregulated in galactose media and rapidly induced during glucose incubation. In set2Δ cells, GAL1 transcript levels were slightly increased at 120 min in galactose media, while HPF1 was constitutively derepressed as previously reported[24]. In contrast, GAL7 and PYK1 showed delayed induction in set2Δ during galactose and glucose incubation, respectively (Supplementary Fig. 1a). Therefore, Set2 plays an important role when gene expression is dynamically changing.

To further explore how Set2 affects gene expression dynamics, total RNA was analysed by high-resolution, strand-specific tiled arrays[25,26]. Importantly, set2Δ did not change mRNA levels of transcriptional regulators, including GAL4, GAL80, MIG1 and TUP1 (Supplementary Fig. 1b). Set2-regulated genes were identified as those showing obvious differences by visual inspection of the tiled array data with at least a twofold increase or decrease in quantitated gene expression levels at one or more time points. In total, 78 genes negatively regulated by Set2 were identified, 60 of which were induced by galactose and 18 that were constitutively expressed (Fig. 1b–d). For the galactose-inducible, Set2-repressed genes, the strongest differences were seen during transition periods (Fig. 1c), explaining why these 60 genes were not previously identified as Set2-regulated[24]. Therefore, H3K36 methylation by Set2 can modulate the rate at which some genes respond to environmental changes.

**Overlapping transcription targets Set2–Rpd3S to promoters.** Visual inspection of the tiled array data revealed that at least 69% of Set2 repressed promoters were overlapped by RNA transcription (non-coding cryptic unstable transcripts (CUTs) or stable unannotated transcripts (SUTs), or possibly read-through transcripts from the adjacent gene) either coming from an upstream or antisense promoter. In contrast, only 12% of yeast genes overlap with lncRNA[27], suggesting that lncRNA transcription and Set2 might cooperate to regulate gene expression.

At AAD10 (Fig. 2a,c), YNR068C, DAL5 (Fig. 3c), SSA3 and ZRT1 (Supplementary Fig. 2a), distal upstream promoters produce a sense direction transcript that apparently continues through the open reading frame. These genes possess a second promoter, much closer to the coding region, which is derepressed by set2Δ. In contrast, the majority of Set2-repressed genes, including SUL1 (Fig. 2a), STL1 and ARO10 (Supplementary Fig. 2a) exhibit overlapping antisense transcription. At these genes the lncRNA transcript levels were unaffected, but target mRNA transcripts were strongly increased in cells lacking SET2.

To investigate the molecular mechanism by which Set2 represses its target genes, histone methylation was analysed using data from Pokholok et al.[11]. Although H3K4me3 and H3K36me3 are typically enriched in promoters and 3′-transcribed regions, respectively, Set2-repressed genes with lncRNA transcription have non-canonical patterns (Fig. 2b). Transcription throughout the YJR154W-AAD10 locus is seen in rich media, yet Aad10 protein is not detected[28,29]. An H3K4me3 peak is seen at the YJR154W promoter but not at AAD10, suggesting that the AAD10 promoter is inactive (Fig. 2b). We suspected that transcription from the YJR154W promoter produces a long RNA that overlaps the AAD10 promoter. To test this idea, we probed an RNA blot with sequences in the AAD10 gene. In wild-type cells grown in raffinose media, a single species consistent with transcription from the YJR154W promoter through the AAD10 gene was detected. In set2Δ cells, an additional short transcript initiating from the AAD10 promoter was observed (Fig. 2c). Confirming the tiled array data, loss of SET2 significantly increased AAD10 but not lncRNA transcription during galactose incubation (Fig. 2c).

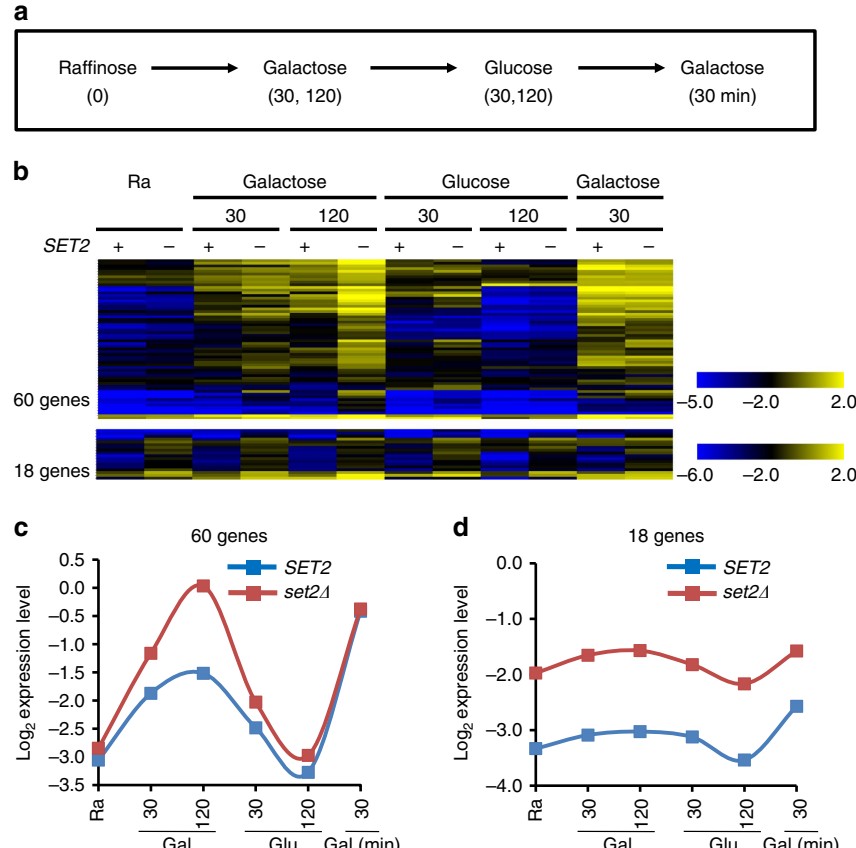

**Figure 1 | Set2 negatively regulates the kinetics of transcriptional induction.** (**a**) Schematic representation of the time course experiments to determine changes in transcript levels upon carbon source shifts. (**b**) RNA samples from the time course experiments in **a** were analysed by high-density tiling arrays. Normalized, log$_2$-transformed mRNA expression levels were visualized with the Multi Experiment Viewer. $+$ and $-$ indicate *SET2* and *set2Δ*, respectively. Set2-repressed genes were identified as those showing at least twofold increase in transcript levels at one or more time points. (**c**) Averaged profile of expression signals of 60 genes from top set in **b**. (**d**) Averaged profile of expression signals of 18 genes from bottom set in **b**. Gal, galactose; Glu, glucose; Ra, raffinose.

Prevention of H3K36 methylation by mutating H3K36 to alanine also increased *AAD10*, further indicating that H3K36 methylation by Set2 directly represses *AAD10* transcription (Fig. 2c). Note that the wild-type control in this experiment carries a single copy of histone H3 and H4 on a plasmid, which may account for the partially derepressed *AAD10* mRNA levels.

The *SUL1* gene is inactive in cells grown in rich media, but has an overlapping antisense transcript, *SUT452*, that is constitutively transcribed (Fig. 2a). Accordingly, H3K4me3 peaks at the 3′-end of *SUL1*, where the *SUT452* promoter is located and H3K36me3 is enriched over the *SUL1* promoter region (Fig. 2b). To test if H3K36 methylation at the *SUL1* and *AAD10* promoters mediate repression through the Rpd3S HDAC, chromatin immunoprecipitation (ChIP) was used to quantitate histone H4 acetylation normalized to total nucleosome levels as measured by histone H3. Relative to the lncRNA promoters, the *AAD10* and *SUL1* promoters had reduced acetylation in wild-type cells (Fig. 2d and Supplementary Fig. 2b). Loss of *SET2* increased acetylation by threefold at the mRNA promoters, but also produced a small increase at the lncRNA promoters. Eaf3 and Rco1, two components of Rpd3S, were also required for deacetylation of the *AAD10* and *SUL1* promoters (Fig. 2d and Supplementary Fig. 2b). Interestingly, acetylation increases were slightly higher in *set2Δ* than in Rpd3S mutants, suggesting additional roles for H3K36 methylation. Previous studies showed that H3K36 methylation also affects binding of a chromatin remodeller ISW1, as well as the histone chaperones, Spt6, Spt16

and Asf1, to chromatin[30,31]. These may also affect histone acetylation levels.

Eaf3 and Rco1 also mediated repression of the Set2-repressed genes *ZRT1*, *STL1* and *SUL1* (Supplementary Fig. 2c). Together, these results support the model that lncRNA transcription can target H3K36 methylation to mRNA promoters and thereby recruit Rpd3S to repress transcription.

**Distance between promoters determines Set2 or Set3 response.** We previously showed that overlapping lncRNA transcription can also target H3K4 dimethylation to mRNA promoters, recruiting the Set3C HDAC to slow the kinetics of transcription induction[25]. Although the Set1–H3K4me2–Set3C and the Set2–H3K36me3–Rpd3S pathways can both be targeted by lncRNA transcription, the microarray data indicates they mostly affect different genes. For example, Set3C represses *DCI1* but not *AAD10*, while Set2–Rpd3S has the opposite pattern (Figs 2c and 3a,b). Similarly, the Set2-repressed genes *YNR068C* and *DAL5* are not repressed by Set3 (Fig. 3c and Supplementary Fig. 3a,b). Comparing the 78 Set2-repressed genes to the 119 genes defined as Set3-repressed, there were 21 in common.

Our models predict that the distance between the lncRNA and mRNA promoters determines whether overlapping lncRNA transcription causes H3K4me2 or H3K36me2/3 to predominate at the mRNA promoter, and in turn this determines response to the two HDAC pathways. Illustrating this correlation, high

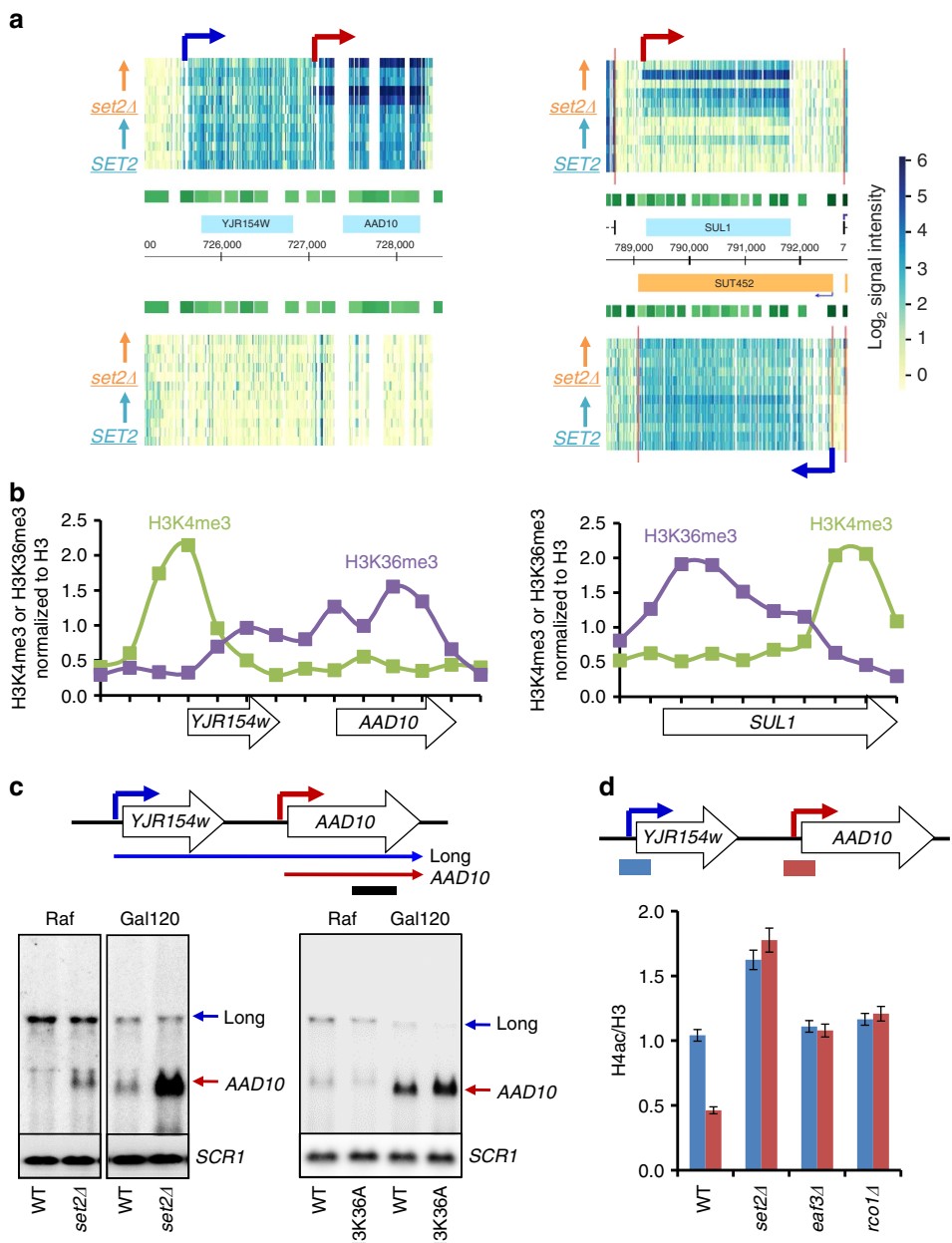

**Figure 2 | Overlapping lncRNA transcription localizes H3K36me3 and Rpd3S to target promoters.** (**a**) *AAD10* has a constitutively active distal promoter (blue arrow) and a proximal promoter (red arrow) induced during galactose incubation in *set2Δ*. *SUL1* has a proximal promoter (red arrow) activated in galactose media in *set2Δ* and a constitutively active antisense promoter (blue arrow). Light blue boxes show open reading frame (ORF) positions and orange box shows lncRNA transcription. Time course as in Fig. 1a is arrayed from bottom to top for the indicated strains. Increased blue colour indicates more transcript hybridizing to the array. (**b**) Histone methylation patterns of *AAD10* and *SUL1* from cells grown in YPD were analysed using the data set from Pokholok *et al.*[11]. Whereas the *AAD10* distal promoter and *SUL1* antisense promoter have high levels of H3K4me3, the core mRNA promoters have high levels of H3K36me3. White boxes show the ORF position. (**c**) Northern blot analysis of *AAD10* transcript with a 3′-strand-specific DNA probe. The indicated cells were grown in synthetic complete (SC) medium containing raffinose (Raf) and shifted to SC-galactose media for 120 min (Gal120). Bottom panels show two transcripts of *AAD10* detected by northern blot analysis, which are schematicized at top. Blue arrow is a distal promoter that produces a lncRNA and red arrow is a proximal promoter for *AAD10* mRNA transcription. A bar underneath upper panel indicates position of probe used for northern blot analysis. (**d**) The Set2–Rpd3S pathway deacetylates histones at the *AAD10* core promoter. Crosslinked chromatin from the indicated strains grown in YPD (where *set2Δ* also derepresses *AAD10*) was precipitated with anti-H3 or anti-acetyl H4 as indicated. PCR analysis of the precipitated DNA was carried out on both distal (blue bars) and proximal (red bars) promoters of *AAD10*. A non-transcribed region near the telomere of chromosome VI was used for an internal control. The signals for acetyl H4 were quantitated and normalized to the total H3 signal, and the ratios were graphed. Error bars show the s.d. calculated from two biological replicates, each with three technical replicates. Similar results were obtained with cells grown in galactose for 120 min. WT, wild type.

H3K36me3 is seen over the Set2-repressed *AAD10*, *YNR068C*, *SUL1* and *DAL5* promoters (Fig. 2b and Supplementary Fig. 3d,e), while the Set3-repressed *DCI1* promoter is within a zone of H3K4me2 (Fig. 3a and Supplementary Fig. 3c). The *DCI1* lncRNA and mRNA promoters are ∼0.55 kb apart, while the *AAD10* promoters are 1.6 kb apart.

To directly test whether altering the distance between the two promoters changes response to Set2–Rpd3S versus Set3C,

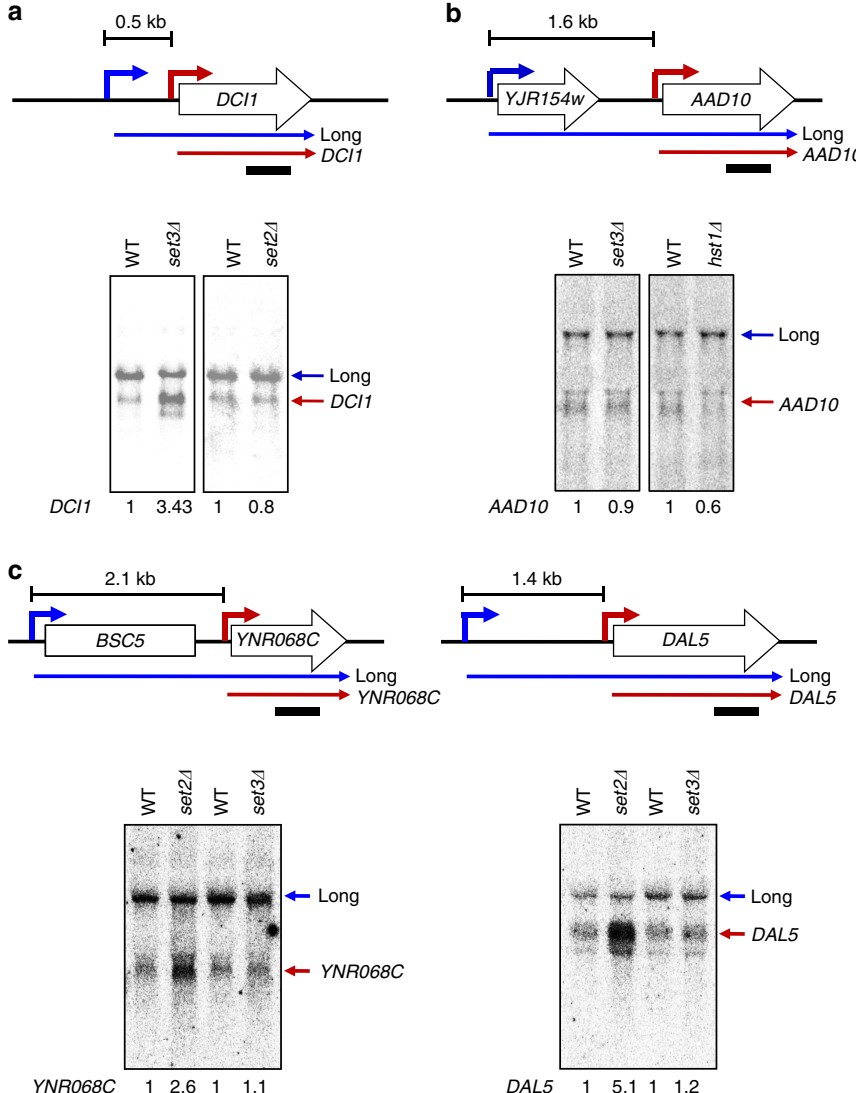

**Figure 3 | Distinct effects of Set2 and Set3 on gene repression. (a)** *DCI1* induction is delayed by Set3 but not by Set2. Cells indicated were grown in synthetic complete (SC) medium containing raffinose and shifted to SC-galactose media for 120 min (Gal120). Bottom panels show two transcripts of *DCI1* detected by northern blot analysis, which are schematicized at top. Blue arrow is a distal promoter that produces a lncRNA and red arrow is a proximal promoter for *DCI1* mRNA transcription. The number between a distal and a lncRNA promoter of upper panel indicates the distance. The bar underneath indicates the probe position used for northern blot analysis. The number underneath of the gel images is the average expression value of *DCI1* from northern blot analyses with two independent RNA samples. **(b)** Set3C does not affect *AAD10* induction. Northern blot analysis of *AAD10* was done as in **a**. **(c)** Northern blot analyses for *YNL068C* and *DAL5* were carried out as in **a**. WT, wild type.

deletions of 500 or 941 bp were made between the upstream and downstream promoters at *AAD10* (Fig. 4a). Northern blot analyses showed the upstream transcript with expected size change was still expressed in raffinose media, although the deletions did cause some reduction in levels or stability (Fig. 4b and Supplementary Fig. 4b). Furthermore, both the full-length and shortened upstream RNA levels decreased during galactose incubation as expected (Fig. 4b and Supplementary Fig. 4a,b).

We next examined the effects on *AAD10* expression. A slight increase was seen in Δ941 cells, but *AAD10* mRNA was induced by galactose similarly in wild-type and Δ500 cells (Fig. 4b). ChIP showed that shortening the distance between the two promoters alters the histone modifications. Whereas H3K36me3 was enriched at the *AAD10* promoter in wild-type cells, significantly lower levels were seen in Δ941 cells (Fig. 4c). Deletion of *SET2* increased histone acetylation at the *AAD10* promoter in wild-type but not in Δ941 cells (Fig. 4d), as expected

if reduced levels of H3K36me3 in Δ941 cells fail to recruit Rpd3S to the *AAD10* promoter.

Northern blot analysis showed that deletion of *SET2* slightly increased basal expression of *AAD10* mRNA in raffinose media (Fig. 4e), and a similar effect was seen on deletion of the gene for the Rpd3S subunit Rco1 (Supplementary Fig. 4c). After 120 min of galactose induction, *AAD10* transcript levels increased. Notably, transcription was strongly increased in either *set2Δ* or *rco1Δ*, confirming repression of the *AAD10* promoter by the Set2–Rpd3S pathway. However, loss of Set2 or Rco1 had much less effect in Δ941 cells (Fig. 4e and Supplementary Fig. 4c). Therefore, shortening the distance between the two promoters changes the methylation and acetylation status of the *AAD10* promoter, and alleviates the repressive effect of Set2–Rpd3S pathway.

To demonstrate that the effect of promoter–promoter distance on Set2-mediated *AAD10* repression is not due to specific intervening

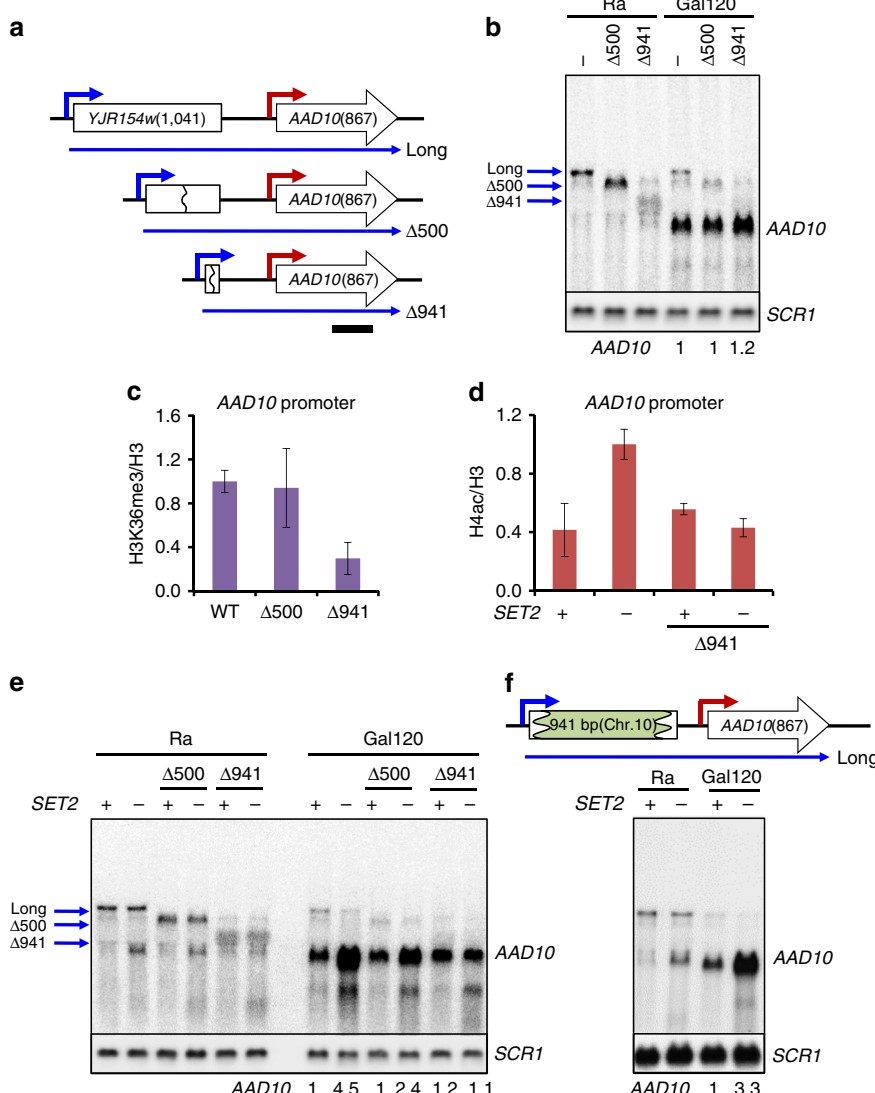

**Figure 4 | Shortening the distance between the two promoters at *AAD10* alleviates Set2–Rpd3S-mediated repression.** (**a**) Schematic representation of *AAD10* locus showing positions of 500 (Δ500) and 941 bp (Δ941) deletions. The upstream and downstream promoters are shown in blue and red, respectively, and the long transcript is shown as blue arrow below. Numbers in parentheses indicate size of the associated open reading frame and black bar indicates the probe position used for northern blot analysis. (**b**) Northern blot analysis of *AAD10* transcripts as in Fig. 3a. Strains from **a** were grown in synthetic complete (SC) medium containing raffinose (Ra) and shifted to SC-galactose media for 120 min (Gal120). *SCR1* is used for a loading control (bottom). (**c**) H3K36me3 at the *AAD10* promoter is significantly decreased in Δ941cells. Crosslinked chromatin from the indicated strains grown in YPD was precipitated with anti-H3 or anti-H3K36me3 as indicated. PCR analysis of the precipitated DNA was carried out on the core promoter of *AAD10*. A non-transcribed region near the telomere of chromosome VI was used for an internal control. The signals for anti-H3K36me3 were quantitated and normalized to the total H3 signal, and the ratios were graphed. Error bars show the s.d. calculated from two biological replicates, each with three technical replicates. (**d**) Set2-targeted histone deacetylation is abrogated in Δ941cells. ChIP assay for anti-acetyl H4 was done as in Fig. 2d. Error bars show the s.d. calculated from two biological replicates, each with three technical replicates. (**e**) Set2 repression of *AAD10* is reduced in Δ941 cells. Northern blot analysis of *AAD10* was done as in **b**. *SCR1* is used as a loading control (bottom). (**f**) Insertion of a 941 bp of DNA from chromosome 10 restores *AAD10* repression by Set2. Northern blot analysis of *AAD10* was done as in **b**. *SCR1* is used as a loading control (bottom). WT, wild type.

sequences, a 941 bp of DNA fragment from a non-transcribed region of chromosome 10 (from 710,600 to 711,540) was inserted between the two promoters in Δ941 cells (designated + 941, Fig. 4f and Supplementary Fig. 4d). While a DNA probe for the chromosome 10 sequence detected the lncRNA transcript only in + 941 cells, a DNA probe from *AAD10* sequence hybridized with lncRNA transcript in both wild-type and + 941 cells (Supplementary Fig. 4d). Although deletion of *SET2* had little effect on *AAD10* repression in Δ941 cells (Fig. 4e), a strong increase of *AAD10* transcript was seen in + 941 cells lacking *SET2* (Fig. 4f). This result clearly indicates that distance, and not specific DNA or

RNA sequences, between the two promoters is critical for the Set2 sensitivity of *AAD10*.

In Δ941 cells, the distance between the upstream promoter and that of *AAD10* is similar to that of the Set3-repressed *DCI1* gene. ChIP experiments showed that H3K4me2 at the *AAD10* promoter was low in wild-type cells, but significantly increased by Δ941 (Fig. 5a). As predicted by our models, an increase in H4 acetylation was seen upon *SET3* deletion in Δ941 cells (Fig. 5b), consistent with H3K4me2 targeting Set3C to the Δ941 *AAD10* promoter. Northern blots were used to determine how the H3K4me2–Set3C pathway affects *AAD10* transcription. In cells grown in raffinose,

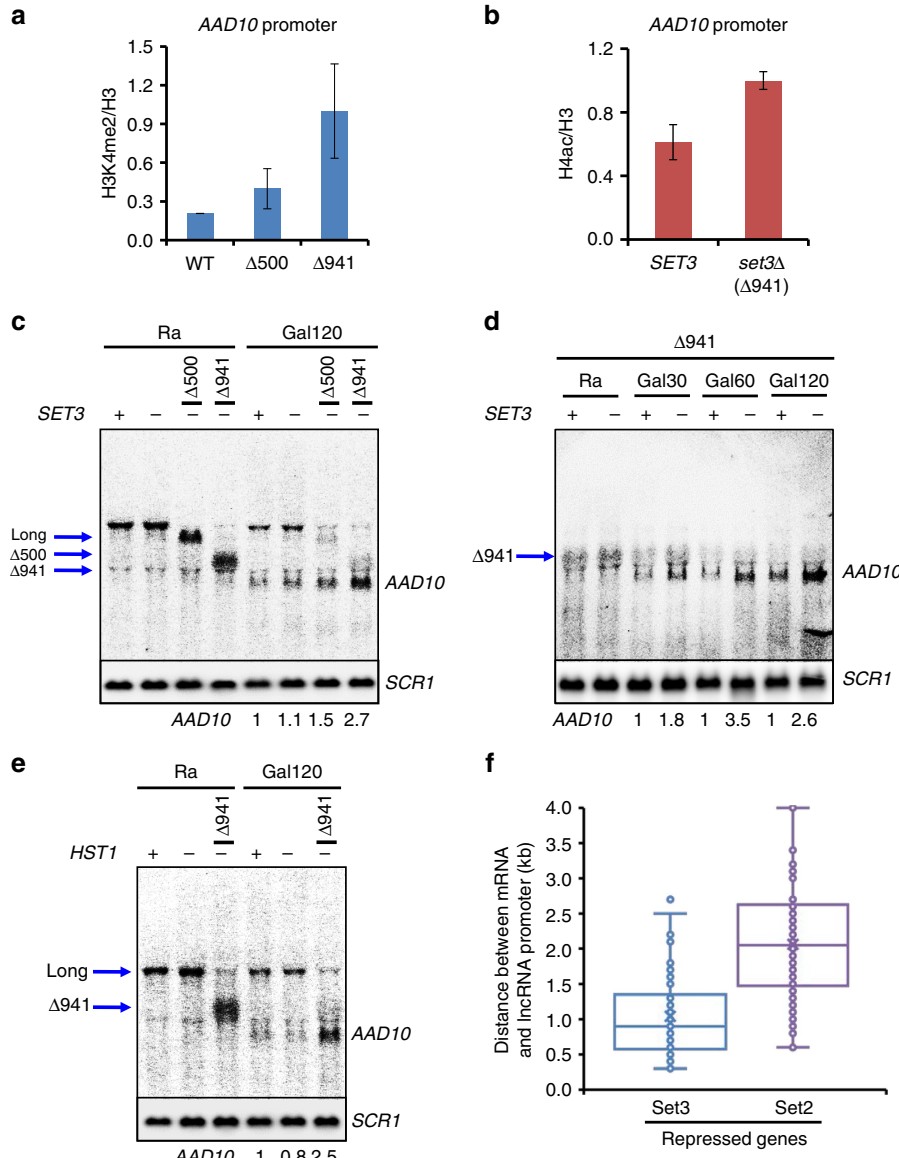

**Figure 5 | Set3C represses *AAD10* transcription in Δ941 cells.** (**a**) Shortening the distance between the two promoters of *AAD10* increases H3K4me2 at the core promoter. ChIP assay for H3K4me2 was done as in Fig. 4c. Error bars show the s.d. calculated from two biological replicates, each with three technical replicates. (**b**) Set3 deacetylates histones at *AAD10* promoter in Δ941 cells. ChIP assay for anti-acetyl H4 was done as in Fig. 2d. Error bars show the s.d. calculated from two biological replicates, each with three technical replicates. (**c**) Loss of Set3 derepresses *AAD10* transcription in Δ941 cells. Figure shows northern blot analysis of the upstream and *AAD10* transcripts. The indicated strains were grown in synthetic complete (SC) medium containing raffinose (Ra) and shifted to SC-galactose media for 120 min (Gal120). *SCR1* is used for a loading control (bottom). (**d**) Set3 delays induction of *AAD10* in Δ941 cells. Northern blot analysis of *AAD10* was done as in **c**. *SCR1* is used as a loading control (bottom). (**e**) Set3C represses *AAD10* transcription in Δ941 cells. Wild-type (WT) and a mutant for Set3C (*hst1Δ*) were analysed by northern blot as in **c**. (**f**) The average distance between mRNA and lncRNA promoters for Set3- or Set2-repressed genes. The distance between the two promoters of individual genes was measured using the tiling array data sets. Circles represent measured distance values, with x showing the mean value. The two boxes and two whiskers represent the four quartiles, with the line between the two boxes showing the median distance value.

loss of Set3C (by deletion of either *SET3* or the Set3C HDAC subunit *HST1*) showed no effect on the upstream transcript and no derepression of *AAD10* (Figs 3b and 5c,e). Furthermore, wild-type cells and Set3C mutants produced similar levels of *AAD10* mRNA upon galactose induction. In contrast, when the promoters are brought closer by Δ941, a 2.7-fold increase of *AAD10* transcript was observed in the absence of Set3C (Fig. 5c), and the induction kinetics were changed (Fig. 5d). Therefore, while Set2–Rpd3S mediates *AAD10* repression in its normal context, shortening the distance between the two promoters abrogates the response to Rpd3S and confers sensitivity to Set3C repression.

We sought to determine whether promoter distance correlates with which pathway mediates gene repression by overlapping transcription. Distances between promoters were plotted for the 59 Set2-repressed genes (this study) and 64 Set3-repressed genes (Kim *et al.*[25]), where a single ncRNA fully overlaps the mRNA promoter. While the average distance between the lncRNA and mRNA promoters of Set3-repressed genes was ∼0.9 kb, the two promoters of Set2-response genes were on average 2.0 kb apart (Fig. 5f). These results support our proposal that distance between mRNA and lncRNA promoter determines whether Set2-Rdp3S or Set3C represses.

**Inducible cryptic promoters repressed by Set2–Rpd3S**. Multiple transcription elongation factors and chromatin regulators, including the Set2–Rpd3S pathway, contribute to suppression of internal cryptic promoters within mRNA genes[16,21–23]. Interestingly, expression of some cryptic promoters is strongly affected by nutritional conditions[22,25]. The tiling array data were analysed to determine whether any of the cryptic promoters repressed by the Set2–Rpd3S pathway respond to the carbon source shifts.

In *set2Δ* cells, 118 internal cryptic promoters generating a short sense transcript and 639 cryptic promoters producing an antisense transcript were identified (Fig. 6a). There were 44 internal promoters that produce divergent cryptic transcripts, as seen for *RAD28* (Fig. 6b). Remarkably, levels of at least half (416) of the Set2-repressed cryptic promoters were regulated by specific carbon sources (Fig. 6a). Representative genes *SLD3*, *TMA108* and *RAD28* are shown in Fig. 6b, none of which are derepressed by deletion of *SET3* (ref. 25; Supplementary Fig. 5). The cryptic transcripts within *SLD3*, *TMA108* and *RAD28* increased in

galactose media, indicating that their internal promoters require some additional activation signal beyond the absence of Set2.

Microarray and northern blots showed at least three cryptic promoters within the *PCA1* gene that are repressed by the Set2–Rpd3S pathway (Fig. 6c). Interestingly, these cryptic promoters differentially respond to galactose. The 5′-cryptic promoter (No. 1) is constitutively active, while the middle cryptic promoter (No. 2) is downregulated and the 3′-promoter (No. 3) is induced during galactose incubation. Furthermore, the 3′-cryptic promoter appears to be bidirectional as antisense transcripts are also increased. These results show that cryptic promoters repressed by the Set2–Rpd3S pathway differentially respond to environmental signals, and it will be interesting to decipher how this regulation is carried out.

## Discussion

The Set2–Rpd3S pathway has been shown to suppress cryptic internal transcription initiation sites, but a role in regulating

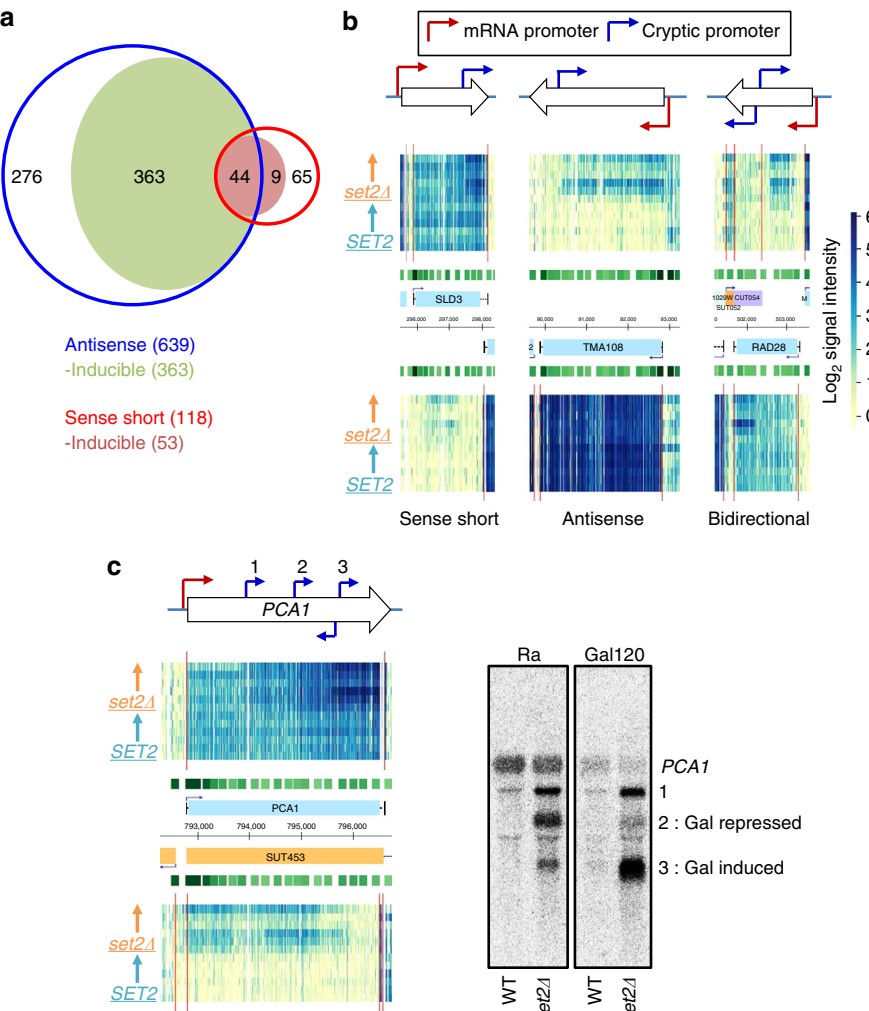

**Figure 6 | Set2 represses inducible cryptic promoters.** (**a**) Tiling array data identified 757 cryptic transcripts increased in *set2Δ*. Venn diagram shows the number of genes with internal cryptic promoters repressed by Set2: 639 antisense and 118 sense direction. Among these, 55% (416 promoters) are more active during carbon source shifts in the absence of *SET2*, indicating that they are inducible cryptic promoters responding to environmental changes. In all, 44 cryptic promoters repressed by Set2 produce both sense and antisense divergent transcripts. (**b**) Examples of distinct cryptic promoter types. Red arrows and blue arrows show the core promoters and cryptic promoters, respectively. Light blue boxes show the open reading frame position. *SLD3* or *TMA108* has an internal cryptic promoter or an antisense promoter that is activated during galactose incubation. *RAD28* gene has internal cryptic promoters that produce short sense and antisense transcripts. (**c**) Northern blot analysis of *PCA1* cryptic transcripts. *SET2* or *set2Δ* cells were grown in synthetic complete (SC) medium containing raffinose (Ra) and shifted to SC-galactose media for 120 min (Gal120). WT, wild type.

mRNA expression has been elusive. Mutants in this pathway have only small effects on steady-state transcript levels[24]. Here we show that Set2–Rpd3S affects the dynamics of gene expression at a subset of promoters, delaying or repressing transcriptional activation upon carbon source shifts (Fig. 1b,c). Although deletion of *SET2* increased basal transcript levels at some genes, differences were generally greatest during carbon source transition periods. Importantly, most Set2-repressed promoters have overlapping lncRNA transcription from either an upstream promoter or downstream antisense promoter. This transcription, independently of the ncRNA itself, deposits H3K36 methylation over the Set2-repressed promoter, leading to histone deacetylation by Rpd3S (Fig. 7a). This targeted deacetylation slows gene induction, presumably creating a requirement for a stronger or longer duration inducing signal. This type of kinetic regulation is likely to be critically important for cells growing in the wild, where they must continuously alter gene expression patterns to adapt to changing environmental conditions.

Several previously described examples of repressive ncRNA expression fit this model. *IME1*, encoding a master regulator of meiosis, is repressed by expression of the lncRNA *IRT1* in *cis*. *IRT1* transcription brings Set2 and Set3C to the *IME1* promoter and blocking *IRT1* transcription alleviates *IME1* repression[32]. *IME4*, another key regulator of meiosis, is also repressed by antisense transcription[33]. In the absence of Set2, this gene was slightly derepressed in galactose-grown cells. Recently, Nguyen *et al.*[34] studied genes differentially expressed in glucose and galactose, and showed that many of these have promoters with overlapping

transcription[34]. For example, *HMS2* transcription negatively impacts the overlapping antisense RNA SUT650 and the downstream gene *BAT2* (ref. 34). Interestingly, we see that *BAT2* and SUT650 are Set2-repressed, suggesting that read-through transcription from *HMS2* targets Set2–Rpd3S to the *BAT2* promoter. Set2 effects are also seen at several other loci studied by Nguyen *et al.*, including *BAG7* and *HIS1/GIP2*. The *ZRT1* gene, with a distal upstream promoter and proximal promoter closer to the open reading frame, is also derepressed in *set2Δ* and mutants for Rpd3S (Supplementary Fig. 2a,c). A previous study showed that insertion of transcription terminators between the distal and proximal promoters derepressed *ZRT1* transcription[35]. The behaviour of all these genes can be explained by overlapping lncRNA transcription targeting H3K36 methylation and Rpd3S deacetylation to the core promoters (Fig. 7a).

Interestingly, we discovered many inducible cryptic promoters repressed by the Set2–Rpd3S pathway that were not identified in earlier studies done under steady-state conditions (Fig. 6a). One previous study found that 1,685 genes had increased acetylation within coding regions in *set2Δ*, but only 56% of these produced detectable internal cryptic transcripts[21]. The remaining genes may lack appropriate basal promoter elements for cryptic transcription. Alternatively, any cryptic transcripts produced at these genes in Set2–Rpd3S mutants may be rapidly degraded by nuclear exosome or other nucleases. For example, antisense transcripts are seen at *STE11* or *CTT1* in a *rco1Δrrp6Δ* strain, but not in *rco1Δ* (ref. 36). Our results suggest one additional possibility; these genes may contain cryptic promoters that are only activated transiently or

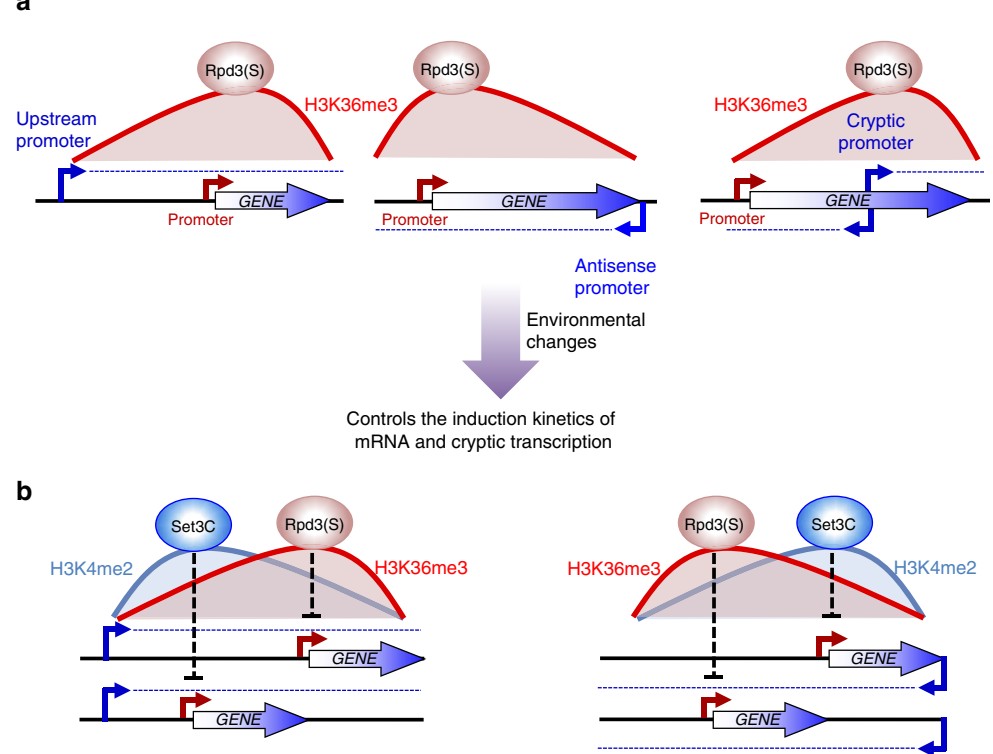

**Figure 7 | Models for regulation of gene expression by Set2–Rpd3S pathway and overlapping lncRNA transcription.** (**a**) At Set2-repressed promoters, transcription from a distal or antisense promoter targets H3K36me3 to the core promoter of mRNA target genes. Rpd3S deacetylates histones in the mRNA promoter region, resulting in delayed or reduced induction. For suppression of internal cryptic promoters, transcription from the mRNA promoter of mRNA gene targets H3K36me3 and Rpd3S deacetylation in 3′-transcribed regions. These cryptic promoters may also respond to environmental changes. (**b**) Position of the lncRNA promoter specifies the repressive effects of two distinct HDACs, Set3C and Rpd3S. H3K4me2 and H3K36me3 are targeted to 5′- and 3′-regions of lncRNA expression, respectively. In 5′-regions, Set3C binds to H3K4me2 to deacetylate histones and repress the mRNA promoter. In contrast, in 3′-regions of lncRNA transcription, Rpd3S binds to H3K36me3 to deacetylate histones and repress mRNA promoters. These mechanisms are also applicable to genes with antisense transcription.

under certain conditions. In our study, ~55% of Set2-repressed cryptic promoters were most active during carbon source shifts (Fig. 6a), indicating these cryptic promoters are responding to environmental signals, as well as to loss of the Set2–Rpd3S pathway. As these responses are often different from the coupled mRNA promoter, it will be interesting to figure out where the relevant regulatory elements reside.

It should be noted that increased internal cryptic initiation may have affected earlier genome-wide microarray analyses that used a limited number of DNA probes, as 3′-probes can hybridize with both full-length mRNA and short cryptic transcripts. For example, PCA1 and STE11 are annotated as Set2-repressed genes based on microarrays[16,21,24]. However, their full-length mRNAs are unaffected in Set2–Rpd3S mutants (Fig. 6c). The increase in signal is due to sense-strand internal cryptic transcripts that are derepressed by a SET2 deletion.

Another important finding of this study is that Set2-mediated effects seen in the first galactose incubation are generally not observed during the second galactose pulse (Fig. 1c). This pattern was also observed for Set3C-mediated delays of gene induction[25]. These results suggest a model in which chromatin regulation by ncRNA transcription could contribute to 'memory' of recent gene expression. Set2-repressed promoters that have not been expressed recently tend to have high levels of H3K36me3 and low acetylation due to the overlapping ncRNA transcription. During the first galactose induction the mRNA promoter marks will be replaced with H3K4me3 and high acetylation. Even if mRNA transcription levels have returned to repressed levels after two additional hours in glucose media, H3K36 methylation may not yet have been re-established over the mRNA promoter (perhaps because the relevant ncRNA promoter has now itself repressed by the overlapping coding RNA transcription). Therefore, the second galactose induction would not need to overcome Set2–Rpd3S repression and the response will be more rapid. In cells lacking Set2 or Rpd3S, the kinetics of the first induction are more similar to the second.

There are obvious parallels between Set2–Rpd3S and Set3C in mediating repression by overlapping ncRNA transcription. It is therefore pertinent to ask whether the two pathways operate independently. Our data indicate that the majority of genes and cryptic transcripts repressed by Set2 are not affected by deletion of SET3, and vice versa. Response appears to correlate with the relative levels of H3K4me2 or H3K36me3, the respective localizing marks for Rpd3S and Set3C HDACs (Fig. 7b). This is most clearly demonstrated by experiments reducing the distance between AAD10 and a repressive upstream promoter. The Δ941 deletion changes the methylation pattern at AAD10, abrogating Set2–Rpd3S repression while making the promoter sensitive to Set3C (Figs 4 and 5). There is a subset of promoters responsive to both pathways, as previously reported for IME1, presumably because of the overlap between H3K4me2 and H3K36me3 at these loci[32].

The roles of cryptic transcription and its ncRNA products are not yet fully understood. Previous studies show that some cryptic transcripts can be translated into short proteins[22,37]. There are also cases where the ncRNA can serves as a scaffold or for targeting bound protein complexes[38]. We have shown in this report and in Kim et al.[25] that overlapping ncRNA transcription, independent of the ncRNA itself, can also target histone modifications to mRNA core promoters to modify transcription responses. Given the plethora of ncRNAs with no clear function, these types of cis-mediated chromatin effects may be more common than previously appreciated.

## Methods

**Yeast strains and plasmids.** Yeast strains and plasmids used in this study are listed in Supplementary Table 1. The time course experiments were done with matched SET2 and set2Δ strains as described in the text and as previously published for Set3 analysis[25]. To generate Δ500, Δ941 and +941 strains in Fig. 4, the delitto perfetto strategy was used[39]. Briefly, for the Δ500 and Δ941 strains the CORE cassette containing KanMX4 and KlURA3 was inserted between the two promoters of AAD10. The resulting strain was transformed with oligonucleotides that delete the upstream regions of AAD10 while also removing the CORE markers. To generate +941 strains, the CORE cassette was inserted between the two promoters in Δ941 cells and replaced with a 941 bp of DNA fragment from a non-transcribed region of chromosome 10.

**Reverse transcription–PCR.** RNA was extracted from cells with hot phenol. First-strand cDNA was prepared with 1 µg total RNA, Superscript II reverse transcriptase (Invitrogen) and gene-specific primers. For Supplementary Fig. 1a, one-fiftieth of the cDNA and 0.6 µCi [α-32P] dATP were used for PCR amplification, which consisted of 60 s at 94 °C, followed by 23–26 cycles (determined to be in the linear range of amplification for each primer pair) of 30 s at 94 °C, 30 s at 55 °C and 45 s at 72 °C, followed by 2 min at 72 °C). PCR signals were quantitated by Fujix Phosphoimager. For Supplementary Fig. 2d, cDNA was analysed by real-time quantitative PCR using SYBR Green Supermix and CFX96 cycler (Bio-Rad).

The sequences of oligonucleotides used in this study are listed in Supplementary Table 2.

**Northern blot analysis.** RNA was isolated from cells with hot phenol and 10 µg total RNA from each sample was separated on an agarose gel, blotted to membranes and hybridized to radioactive probes using standard techniques[40]. The sequences of oligonucleotides used for northern blot analysis are listed in Supplementary Table 2. Strand-specific probes were generated by unidirectional PCR in the presence of [α-32P] dATP with only one primer. Hybridization was done in a buffer containing 1% BSA, 7% SDS, 1 mM EDTA (pH 8.0) and 300 mM sodium phosphate buffer (pH 7.2).

**Chromatin immunoprecipitations.** ChIPs were done as previously described with the modifications described below[41]. The following histone antibodies were used: 1 µl of anti-H3K4me2 (Upstate 07-030); 0.5 µl of anti-H3K36me3 (Abcam 9050); 0.5 µl of anti-acetyl H4 (Upstate 06-598); and 2 µl of anti-H3 (Abcam 1791). All antibodies were bound to Protein A-agarose beads and used for precipitation of formaldehyde-crosslinked chromatin. For anti-H3 antibody, binding was done in FA lysis buffer containing 275 mM NaCl. For other antibodies, binding was done in FA lysis buffer containing 1 M NaCl. Precipitated DNAs were analysed in real-time using SYBR Green Supermix and CFX96 cycler (Bio-Rad). Oligonucleotides for PCR analysis are listed in Supplementary Table 2.

**High-resolution tiling arrays.** High-resolution tiling arrays were done as previously described[25]. Briefly, RNA isolated from cells was converted to cDNA with a combination of random and oligo dT priming, labelled and hybridized to tiled Affymetrix arrays carrying the yeast genome. Data are plotted as in[29] and can be viewed by web browser at http://steinmetzlab.embl.de/buratowski_lab/index.html. The raw microarray data can be downloaded from ArrayExpress with the accession code E-MTAP-4268.

**Data availability.** Microarry data sets that support the findings of this study have been deposited in the ArrayExpress repository (https://www.ebi.ac.uk/arrayexpress/) with the accession code E-MTAP-4268.

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

## Acknowledgements

We are grateful to Jerry Workman and Swami Venkatesh (Stowers Institute for Medical Research) for communicating unpublished results, to Sandra Clauder-Münster and the EMBL Genomics Core Facility for technical support, and all members of the Buratowski and Kim labs for helpful advice and discussions. This research was supported by grants to T.K. (Ewha Womans University Research Grant of 2013, the T.J. Park Science Fellowship of POSCO T.J. Park Foundation, the Basic Science Research Program of the National Research Foundation of Korea (NRF) funded by the Ministry of Education, Science and Technology (NRF-2013R1A1A1008634 and NRF-2012R1A5A1048236), the National Research Foundation of Korea funded by Korean Government (NRF-2013S1A2A2035342)), to L.M.S. (Deutsche Forschungsgemeinschaft and US National Institutes of Health GM068717) and to S.B. (US National Institutes of Health GM46498).

## Author contributions

T.K. and S.B. designed the project; J.H.K., B.B.L. and T.K. performed most of the data analyses and experiments; Y.M.O. contributed to the data analyses and gene expression analysis; C.Z. and L.M.S. performed high-resolution tiling array; Y.L. and W.K.K. analysed the data; S.B.L. contributed to designing the experiment in Fig. 4f; T.K. and S.B. wrote the manuscript. All authors discussed the results and commented on the manuscript.

## Additional information

**Competing financial interests:** The authors declare no competing financial interests.

DOI: 10.1038/ncomms16122    OPEN

# Erratum: Modulation of mRNA and lncRNA expression dynamics by the Set2–Rpd3S pathway

Ji Hyun Kim, Bo Bae Lee, Young Mi Oh, Chenchen Zhu, Lars M. Steinmetz, Yookyeong Lee, Wan Kyu Kim, Sung Bae Lee, Stephen Buratowski & TaeSoo Kim

Nature Communications 7:13534 doi: 10.1038/ncomms13534 (2016); Published 28 Nov 2016; Updated 29 Nov 2017

The ArrayExpress accession code is incorrect in this Article. The correct accession code is E-MTAB-4268.

