## [Peer Review File · Nature Communications]

Reviewers' comments:

Reviewer #1 (Remarks to the Author):

Kim et al. investigate the impact of the H3 K36 methyltransferase Set2 on coding and noncoding transcription in yeast using strand-specific tiled arrays. Although Set2 has an established role in repressing cryptic initiation within genes through recruitment of Rpd3S, its impact on mRNA synthesis had not been fully explored. The authors investigate this question and expand upon it by probing transcriptional changes that occur in wild type and set2 mutants in response to a series of carbon source shifts. This protocol elevated the significance of the work by showing that some Set2-dependent effects on transcription are only evident when cells are exposed to changing growth conditions. The major findings of this work are a complete description of Set2-dependent coding and noncoding transcripts in dynamically changing conditions, insights into how noncoding transcription controls mRNA transcription through two distinct histone modification pathways (H3K36me3-mediated recruitment of Rpd3S or H3K4me2-dependent recruitment of Set3C), and identification of novel cryptic transcription events in set2 mutants that are only revealed in a specific growth condition.

The study as a whole is nicely done, both in terms of the genomic analysis and the mechanistic follow-up which takes the work beyond a descriptive phase. The authors describe and test specific hypotheses to reveal the mechanisms controlling whether an mRNA is subject to Set2- or Set3-mediated repression. They conclusively show that the distance between the start sites for the mRNA and a noncoding RNA (e.g. positioned upstream in cis) determines whether the promoter for the mRNA is methylated at K36 or K4, and hence whether repression of the promoter relies on the Rpd3S or Set3C HDACs. Another strength of the work is inclusion of ChIP data and northern analyses to validate hypotheses or confirm the genomic data. Overall, the results provide a description of how deleting SET2 deregulates expression of mRNAs in addition to a well-supported mechanistic explanation of the effects.

Specific comments:

1. The strains used for this study contain intact RNA degradation systems, such as Xrn1 or the nuclear exosome. It is possible that the authors' results underestimate the effect of Set2 on repressing cryptic transcription. I would not recommend repeating the full time course in an rrp6 mutant, since the authors have already tested many growth conditions. However, the authors should address this limitation of their study, either in the text or possibly by testing a small number of conditions.

2. Figure 4E, 4F, 5C and 5D. In these northern blots, the authors test if the distance between the upstream and downstream start sites determines whether the downstream promoter (AAD10 gene) is repressed by a Set2 or Set3-dependent process. The approach is to delete intervening sequences between the start sites. Missing from these blots are controls showing the level of AAD10 mRNA in cells that contain Set2 (or Rco1 or Set3 or Hst1) and have deletions in the AAD10 locus. It is difficult to know how much or little derepression occurs upon deletion of the chromatin factors in the distance mutants, when the appropriate wild type control is shown on a different blot. Since the extent of repression by Set2 (or Rco1 or Set3 or Hst1) is the important point, RNA from strains that contain these factors should be shown on the same blot in parallel with RNA from strains that lack these factors.

3. How many replicas were performed for the northern blots? The authors should provide quantitation.

4. The discovery of new cryptic initiation events in the set2 mutant in different growth conditions is interesting. Perhaps more interesting would be the identification of cryptic initiation events in the wild type strain, as these would indicate that Set2 or some other aspect of chromatin structure responds to changes in the environment. Did the authors see evidence for cryptic initiation in any of their growth conditions in wild type cells?

5. Supplemental Figure 1. Panel A does not add much to the paper- in fact, without quantitation, it's hard to see some of the effects described in the text. Also, the data are confusing. Why should GAL1 and GAL7 behave with such different induction kinetics? Why is the time course for GAL1 induction so slow? Does the persistence of the GAL transcripts in glucose reflect what is known about glucose-repression of these genes? In Panel B, and elsewhere, it should be clearly stated what the individual rows represent.

6. Supplemental Figure 4. The SCR1 loading control in Panel A does not match the AAD10 blot (5 AAD10 samples vs. 10 SCR1 samples).

Reviewer #2 (Remarks to the Author):

The paper by Kim et al. aims at further exploring the role of Set2 and H3K36 methylation in gene regulation through the recruitment of Rpd3S histone deacetylase. The basis of this work is a comparative analysis of transcription by tiling arrays of wild-type and Dset2 cells undergoing a series of carbon source shifts. They find that 18 genes are upregulated in the absence of Set2 and another 59 which activation kinetics is increased in Dset2 following a shift from raffinose to

galactose. Moreover, they find that most of these gene promoters overlap with lncRNA transcription, either in the sense or antisense orientation, and propose that Set2 dependent H3K36me3 deposited during non-coding transcription mediates the recruitment of the HDAC Rpd3S responsible for promoter repression. The same authors have proposed earlier that H3K4me2 deposited by non-coding transcription is also involved in promoter repression by promoting the recruitment of the HDAC Set3.

The experiments presented in this paper support, at least in part, that promoter repression by either the Set3 or Rpd3S dependent pathway depends on how far from the promoter non-coding transcription is initiated. The authors perform a series of experiments on few genes, but mainly on one (AAD10), showing quite convincingly that when upstream non-coding transcription starts more than 1kb from the promoter, repression will mainly depend on H3K36me3 and Set2, but becomes dependent on H3K4me2 and Set3 if this distance is reduced.

The study is well done and provides further insight into how non-coding transcription may contribute to the regulation of gene expression, although Set2 has modest effects, modulating the expression or inducibility of 80 only genes. There are nevertheless few weaknesses in this work that should be addressed in order to strengthen the general conclusions and to make this study acceptable for publication in Nature Communications :

Figure 2C major comment : the Northern blot shows that Set2 negatively affects AAD10 expression when shifting cells from Raf to Gal. To show that the enhanced AAD10 activation in Dset2 is indeed due to the absence of H3K36me3, the same experiment should be performed in a H3K36A mutant.

Minor comment : this Northern blot lacks a loading control. However, since this blot is part of the same blot as shown in Figure 4E, the authors could mention that Figure 2D corresponds to a part of Figure 4E on which RNA loading has been controlled.

Figure 2D : this figure is not satisfactory. Indeed it examines acetylation levels at AAD10 and upstream locus in YPD medium and is supposed to complement Figure 2C comparing AAD10 expression in Raf versus Gal. In order to establish a parallel between

Figure 2C and 2D, the authors should show a Northern blot of WT and Dset2 cells grown in YPD to define whether AAD10 is increased in Dset2 in this medium. Alternatively, they should perform the ChIP of H4Ac at the promoter of YJR154w and AAD10 in WT or mutant cells grown in galactose.

Another unexpected observation in Figure 2D is why the level of H4Ac in Deaf3 and Drco1 does not increase as much as in Dset2. Is another HDAC implicated, and if yes, which one ? Could Rpd3L be involved, since this HDAC was proposed earlier by the authors to act at promoters. This point should be discussed.

Figures 3 and 4 address the importance of the distance between the site of lncRNA initiation and the affected promoter in defining the repression pathway (Set2-H3K36me3/Rpd3 vs Set1-H3K4me/Set3). Figure 3 shows Northern blot analysis of the expression of three Set2-dependent (AAD10, YNR068, DAL5) and one Set3-dependent (DCI1) gene, all regulated by an upstream lncRNA, in the presence or absence of these enzymatic activities (Figure 3) ; the distances between promoters is not always indicated and should be added on the figure. Figure 4 more directly and convincingly addresses the importance of the distance for pathway determination, by creating increasing deletions upstream of the AAD10 gene.

However, what is missing to introduce Figures 3 and 4 is a comparative analysis of the whole set of Set2 or Set3 affected genes to show that more generally the distance between lncRNA and affected promoter is indeed larger in the case of Set2 compared to Set3 repressed genes, and that H3K36me levels are higher at Set2/Rpd3 vs Set1/Set3 repressed promoters and the opposite for H3K4me2. Without such metagene analyses, the proposed model cannot be generalized.

Figure 4B : the long upper band is persisting in the deletion mutants grown in Galactose. Is this a cross-reacting band ? To further implicate the upstream lncRNA in AAD10 repression, the authors should also abolish its expression by mutating its promoter without shortening the YJR154w sequence.

Figures 4C and D examine H3K36me3 and H4Ac at the AAD10 gene in WT or deleted versions of the upstream lncRNA (D500 and D941). As already mentioned for Figure 2B, these ChIP experiments have been performed on extracts from cells grown in YPD and not under galactose inducing conditions as used in the Northern blot of Figure 4B. To be able to compare Figures 4C and 4D to 4B, the authors should either provide a Northern blot with RNA from cells grown in YPD or repeat the histone modification ChIPs using extracts from cells grown in Raf and induced in Galactose.

Moreover, to confirm that the promoter deletions do not affect transcription, it would be good to compare RNA PolII ChIP over the WT and mutant versions, as well as to show that the H3K36me peak is shifted towards the 3'end in the deletion mutants.

Suppl. Figure 2B : same comment as for Figure 2D. The level of H4Ac at the SUL1 gene regulated by an antisense lncRNA in presence or absence of Set2 or Rpd3S components should be measured following a shift from Raf to Gal and not in YPD.

Suppl. Figure 2D does not state in what type of medium the cells were grown before transcript level analysis.

Suppl. Figure 2C legend is missing and the text does not refer to this figure.

Reviewer #3 (Remarks to the Author):

NCOMMS-16-09228

In this manuscript, the authors investigated how the Set2/Rpd3 pathway affects lncRNA transcription in budding yeast upon carbon source shifts. They identified novel lncRNAs, and proposed an interesting model to explain how the distance between lncRNA promoters and other promoters dictates the effects of histone modifying enzymes on lncRNA transcription levels. This model is novel, and has the potential to help unravel the basis for the complex effects of histone modifications on lncRNA transcription. I recommend accepting this manuscript once the following points are addressed.

1. Nguyen et al. (eLife, 03635, 2014) recently reported dynamic changes in both the mRNA and lncRNA transcriptomes of budding yeast upon carbon source shifts. The authors need to compare their data with those from Nguyen et al.
2. The authors used histone modification patterns as evidence for targeting of histone modifying enzymes. This is a very indirect readout. Given that the targeting patterns of the enzymes are central to the authors' model, more direct evidence for enzyme targeting (modifying enzyme ChIP) needs to be provided. This is particularly important as the transcriptional response to mutations, which can also affect histone modifications, is highly complex.
3. Page9, lines 1-2: I believe the authors' statement, "The distance between the lncRNA and mRNA promoters will determine which histone methylation predominates at the RNA promoter" is their model, rather than an established fact. However, this sentence reads as if this were a widely accepted fact. The authors need to modify the sentence.

We thank the reviewers for their insightful comments on our manuscript and we have addressed them accordingly. We hope that the reviewers will now find the revised manuscript ready for publication in Nature Communications.

Reviewer #1 (Remarks to the Author):

Kim et al. investigate the impact of the H3 K36 methyltransferase Set2 on coding and noncoding transcription in yeast using strand-specific tiled arrays. Although Set2 has an established role in repressing cryptic initiation within genes through recruitment of Rpd3S, its impact on mRNA synthesis had not been fully explored. The authors investigate this question and expand upon it by probing transcriptional changes that occur in wild type and set2 mutants in response to a series of carbon source shifts. This protocol elevated the significance of the work by showing that some Set2-dependent effects on transcription are only evident when cells are exposed to changing growth conditions. The major findings of this work are a complete description of Set2-dependent coding and noncoding transcripts in dynamically changing conditions, insights into how noncoding transcription controls mRNA transcription through two distinct histone modification pathways (H3K36me3-mediated recruitment of Rpd3S or H3K4me2-dependent recruitment of Set3C), and identification of novel cryptic transcription events in set2 mutants that are only revealed in a specific growth condition.

The study as a whole is nicely done, both in terms of the genomic analysis and the mechanistic follow-up which takes the work beyond a descriptive phase. The authors describe and test specific hypotheses to reveal the mechanisms controlling whether an mRNA is subject to Set2- or Set3-mediated repression. They conclusively show that the distance between the start sites for the mRNA and a noncoding RNA (e.g. positioned upstream in cis) determines whether the promoter for the mRNA is methylated at K36 or K4, and hence whether repression of the promoter relies on the Rpd3S or Set3C HDACs. Another strength of the work is inclusion of ChIP data and northern analyses to validate hypotheses or confirm the genomic data. Overall, the results provide a description of how deleting SET2 deregulates expression of mRNAs in addition to a well-supported mechanistic explanation of the effects.

We thank the reviewer for the positive remarks on our manuscript.

Specific comments:

1. The strains used for this study contain intact RNA degradation systems, such as Xrn1 or the nuclear exosome. It is possible that the authors' results underestimate the effect of Set2 on repressing cryptic transcription. I would not recommend repeating the full time course in an rrp6 mutant, since the authors have already tested many growth conditions. However, the authors should address this limitation of their study, either in the text or possibly by testing a small number of conditions.

We completely agree with the reviewer. A previous study (Tan-Wang et al., Science, 2012) analyzed genome-wide transcripts in rrp6Δ and rrp6Δrco1Δ in a steady state condition (Rco1 is a subunit of Rpd3S). This study uncovered cryptic transcripts that are detected only in double mutants. One example is an antisense transcript from STE11. This gene produces short sense transcripts in mutants for Set2/Rpd3S but an antisense transcript is only detected in rrp6Δrco1Δ indicating that antisense transcript is rapidly degraded by nuclear exosome. We have added a discussion of this point in discussion section (page 14).

2. Figure 4E, 4F, 5C and 5D. In these northern blots, the authors test if the distance between the upstream and downstream start sites determines whether the downstream promoter (*AAD10* gene) is repressed by a *Set2* or *Set3*-dependent process. The approach is to delete intervening sequences between the start sites. Missing from these blots are controls showing the level of *AAD10* mRNA in cells that contain *Set2* (or *Rco1* or *Set3* or *Hst1*) and have deletions in the *AAD10* locus. It is difficult to know how much or little derepression occurs upon deletion of the chromatin factors in the distance mutants, when the appropriate wild type control is shown on a different blot. Since the extent of repression by *Set2* (or *Rco1* or *Set3* or *Hst1*) is the important point, RNA from strains that contain these factors should be shown on the same blot in parallel with RNA from strains that lack these factors.

As requested, we have repeated northern blot analyses with appropriate controls and new data were now added in Figure 4e and others. The results support the previous ones indicating that *Set2/Rpd3S* has very little effect on *AAD10* repression in $\Delta 941$ cells. We also performed northern blot analysis with $\Delta 941$ (WT) and $\Delta 941$ (*set3* Δ) to directly determine the effect of *SET3* deletion in $\Delta 941$ cells. This result is now added in Figure 5d.

3. How many replicas were performed for the northern blots? The authors should provide quantitation. All northern blot data were from two independent RNA samples and no significant differences were seen. We quantitated the results using Image J and added the quantitation of key results to the figures.

4. The discovery of new cryptic initiation events in the *set2* mutant in different growth conditions is interesting. Perhaps more interesting would be the identification of cryptic initiation events in the wild type strain, as these would indicate that *Set2* or some other aspect of chromatin structure responds to changes in the environment. Did the authors see evidence for cryptic initiation in any of their growth conditions in wild type cells?

Yes we did. We did not systematically analyze these events (their low levels make them difficult to detect), but interested parties will be able to use our array data to find them. One example is SUT650, an internal antisense ncRNA that is increased in galactose incubation. Fred Winston's group showed that some cryptic transcripts, including *FLO8* and *SPB4*, were induced when wild-type cells were shifted from YPD to SC media (Cheung et al., PLOS BIOLOGY, 2008). We also showed that noncoding transcripts including *PAH1* cryptic transcript were induced in WT cells during galactose incubation (Kim et al., Cell, 2012). Therefore, the use of the word "cryptic" can obscure the fact that at least some of these transcripts exist in WT cells.

5. Supplemental Figure 1. Panel A does not add much to the paper- in fact, without quantitation, it's hard to see some of the effects described in the text. Also, the data are confusing. Why should *GAL1* and *GAL7* behave with such different induction kinetics? Why is the time course for *GAL1* induction so slow? Does the persistence of the *GAL* transcripts in glucose reflect what is known about glucose-repression of these genes?

As the reviewer noticed, the induction kinetics of *GAL1* and *GAL10* are very similar, but *GAL7* comes up faster. Based on our models, we believe the difference in the induction kinetics is because *GAL1*

and *GAL10* overlap with at least two noncoding RNA transcriptions that might delay induction. However, there is no overlapping ncRNA transcription at *GAL7*. Previous studies showed that abrogation of the lncRNA transcription at the *GAL1-GAL10* locus promotes faster induction of these genes (Houseley et al., Molecular Cell, 2008; Pinskaya et al., EMBO J, 2009).

In Panel B, and elsewhere, it should be clearly stated what the individual rows represent.

Thank you for point out this omission. The time course schematic is now added on the top and the figure legend now explains how this relates to the rows in the expression array data.

6. *Supplemental Figure 4. The *SCR1* loading control in Panel A does not match the *AAD10* blot (5 *AAD10* samples vs. 10 *SCR1* samples).*

We have corrected this issue.

Reviewer #2 (Remarks to the Author):

*The paper by Kim et al. aims at further exploring the role of Set2 and H3K36 methylation in gene regulation through the recruitment of Rpd3S histone deacetylase. The basis of this work is a comparative analysis of transcription by tiling arrays of wild-type and Dset2 cells undergoing a series of carbon source shifts. They find that 18 genes are upregulated in the absence of Set2 and another 59 which activation kinetics is increased in Dset2 following a shift from raffinose to galactose. Moreover, they find that most of these gene promoters overlap with lncRNA transcription, either in the sense or antisense orientation, and propose that Set2 dependent H3K36me3 deposited during non-coding transcription mediates the recruitment of the HDAC Rpd3S responsible for promoter repression. The same authors have proposed earlier that H3K4me2 deposited by non-coding transcription is also involved in promoter repression by promoting the recruitment of the HDAC Set3. The experiments presented in this paper support, at least in part, that promoter repression by either the Set3 or Rpd3S dependent pathway depends on how far from the promoter non-coding transcription is initiated. The authors perform a series of experiments on few genes, but mainly on one (*AAD10*), showing quite convincingly that when upstream non-coding transcription starts more than 1kb from the promoter, repression will mainly depend on H3K36me3 and Set2, but becomes dependent on H3K4me2 and Set3 if this distance is reduced.*

The study is well done and provides further insight into how non-coding transcription may contribute to the regulation of gene expression, although Set2 has modest effects, modulating the expression or inducibility of 80 only genes. There are nevertheless few weaknesses in this work that should be addressed in order to strengthen the general conclusions and to make this study acceptable for publication in Nature Communications :

We thank the reviewer for the positive remarks on our manuscript.

*Figure 2C major comment : the Northern blot shows that Set2 negatively affects *AAD10* expression when shifting cells from Raf to Gal. To show that the enhanced *AAD10* activation in Dset2 is indeed due to the absence of H3K36me3, the same experiment should be performed in a H3K36A mutant.*

As requested, we performed northern blot analysis with WT and H3K36A mutant and added the new data in **Figure 2c**. Although we saw a higher basal level of *AAD10* transcript in the WT control (this

may be because the strains used here have only one copy of histone H3 and H4 on a plasmid instead of the two in the genome), the mutation of H3K36 to A further increased AAD10 transcript.

Minor comment : this Northern blot lacks a loading control. However, since this blot is part of the same blot as shown in Figure 4E, the authors could mention that Figure 2D corresponds to a part of Figure 4E on which RNA loading has been controlled.

We have added new data with SCR control.

Figure 2D : this figure is not satisfactory. Indeed it examines acetylation levels at AAD10 and upstream locus in YPD medium and is supposed to complement Figure 2C comparing AAD10 expression in Raf versus Gal. In order to establish a parallel between Figure 2C and 2D, the authors should show a Northern blot of WT and Dset2 cells grown in YPD to define whether AAD10 is increased in Dset2 in this medium. Alternatively, they should perform the ChIP of H4Ac at the promoter of YJR154w and AAD10 in WT or mutant cells grown in galactose.

As requested, to allow direct comparisons we performed a northern blot of WT and set2Δ grown in YPD and ChIP of H4ac in Gal 120min (see below). AAD10 derepression by set2Δ was very similar to that in Raffinose. Furthermore, H4ac was still increased in cells grown at Gal120min. Since these results did not provide additional information, these are not included in the figures but were instead noted in the Fig 2D legend.

Another unexpected observation in Figure 2D is why the level of H4Ac in Deaf3 and Drco1 does not increase as much as in Dset2. Is another HDAC implicated, and if yes, which one ? Could Rpd3L be involved, since this HDAC was proposed earlier by the authors to act at promoters. This point should be discussed.

Papers from the Workman group showed that H3K36me3 by Set2 was also required for ISWI binding to chromatin (Smolle et al., NSMB, 2012). Furthermore, this modification also affected chromatin binding of histone chaperones, Spt6, Spt16, and Asf1 (Venkatesh et al., Nature, 2012). All these factors functionally interact with Set2 and might affect histone acetylation levels. We have now added this point to the text.

We can't rule it out, but we don't favor that Rpd3L contributes to Set2-mediated histone deacetylation because deletion of Rpd3L specific subunit Pho23 increases acetylation at the promoter but not further downstream where the overlapping lncRNAs initiate (Kim and Buratowski, Cell, 2009).

Figures 3 and 4 address the importance of the distance between the site of lncRNA initiation and the affected promoter in defining the repression pathway (Set2-H3K36me3/Rpd3 vs Set1-H3K4me/Set3). Figure 3 shows Northern blot analysis of the expression of three Set2-dependent (AAD10, YNR068, DAL5) and one Set3-dependent (DC11) gene, all regulated by an upstream lncRNA, in the presence or absence of these enzymatic activities (Figure 3) ; the distances between promoters is not always indicated and should be added on the figure.

Good point. The distances between the two promoters have been added in Figure 3.

Figure 4 more directly and convincingly addresses the importance of the distance for pathway determination, by creating increasing deletions upstream of the AAD10 gene.

However, what is missing to introduce Figures 3 and 4 is a comparative analysis of the whole set of Set2 or Set3 affected genes to show that more generally the distance between lncRNA and affected promoter is indeed larger in the case of Set2 compared to Set3 repressed genes, and that H3K36me levels are higher at Set2/Rpd3 vs Set1/Set3 repressed promoters and the opposite for H3K4me2. Without such metagene analyses, the proposed model cannot be generalized.

We thank the reviewer for this excellent suggestion. We measured the distance between the two promoters of Set2- or Set3-repressed genes using our tiling array datasets. The data is now added in Figure 5f. The median distance between the two promoters of Set3 response genes was about 0.9kb, while the two promoters of Set2-repressed genes were 2.0kb apart. While there is some overlap, the distributions of distances strongly support our model.

Figure 4B : the long upper band is persisting in the deletion mutants grown in Galactose. Is this a cross-reacting band ?

As the reviewer pointed out, the long upper band is likely a cross-reacting one because this was not detected when we used a DNA probe from the upstream region (Supplementary Figure 4b). This region of the gel is where the ribosomal RNA runs and it appears this may be contributing to the background.

To further implicate the upstream lncRNA in AAD10 repression, the authors should also abolish its expression by mutating its promoter without shortening the YJR154w sequence.

This is a good experiment and we tried very hard to get it to work. We first tried to delete the promoter sequence of the YJR154w, and subsequently inserted the CYC1 terminator between the two promoters. Unfortunately, in both cases the mutation failed to abolish the upstream transcription. We're not sure exactly where these upstream transcripts are initiating, but obviously we can't make any conclusions.

Although we're not able to do this experiment for Aad10, we note that previous reports showed that loss of lncRNA transcription derepresses IME4 and ZRT1, two target genes of Set2 identified in this study. We included this point in the discussion section (Hongay et al., Cell, 2006; Toesca et al., PLOS Genet., 2011). ZRT1 had an upstream lncRNA like AAD10 and insertion of ADH1 terminator between the upstream promoter and mRNA promoter increased ZRT1 transcript (Toesca et al., PLOS Genet., 2011). These previous reports support our models.

In lieu of the suggested experiment, our revised manuscript contains a new experiment showing that insertion of a random 941bp DNA fragment (from a nontranscribed region on a different chromosome)

between the two promoters in \$\Delta 941\$ cells restored Set2-mediated repression of AAD10. This result argues that lncRNA transcription, but not RNA or DNA sequence, directly mediates AAD10 repression by Set2.

Figures 4C and D examine H3K36me3 and H4Ac at the AAD10 gene in WT or deleted versions of the upstream lncRNA (D500 and D941). As already mentioned for Figure 2B, these ChIP experiments have been performed on extracts from cells grown in YPD and not under galactose inducing conditions as used in the Northern blot of Figure 4B. To be able to compare Figures 4C and 4D to 4B, the authors should either provide a Northern blot with RNA from cells grown in YPD or repeat the histone modification ChIPs using extracts from cells grown in Raf and induced in Galactose.

We have performed a northern blot with RNA from cells grown in YPD and ChIP for histone acetylation at Gal120. Please see the data attached on Page 4.

Moreover, to confirm that the promoter deletions do not affect transcription, it would be good to compare RNA PolII ChIP over the WT and mutant versions, as well as to show that the H3K36me peak is shifted towards the 3'end in the deletion mutants.

We have performed RNA PolII ChIP in wild type and promoter deletion mutants. Since the lncRNA transcript is stably detected in YPD or raffinose, Rpb3 crosslinking was determined in YPD. Overall, Rpb3 crosslinking was similar in WT, \$\Delta 500\$, and \$\Delta 941\$ strains. We chose not to add this to the paper, but can do so if the editor thinks it important.

Suppl. Figure 2B : same comment as for Figure 2D. The level of H4Ac at the SUL1 gene regulated by an antisense lncRNA in presence or absence of Set2 or Rpd3S components should be measured following a shift from Raf to Gal and not in YPD.

Please see the SUL1 ChIP in the response for Figure 2D.

Suppl. Figure 2D does not state in what type of medium the cells were grown before transcript level analysis.

This has been corrected.

Suppl. Figure 2C legend is missing and the text does not refer to this figure.

This figure has been removed.

Reviewer #3 (Remarks to the Author):

NCOMMS-16-09228

In this manuscript, the authors investigated how the Set2/Rpd3 pathway affects lncRNA transcription in budding yeast upon carbon source shifts. They identified novel lncRNAs, and proposed an interesting model to explain how the distance between lncRNA promoters and other promoters dictates the effects of histone modifying enzymes on lncRNA transcription levels. This model is novel, and has the potential to help unravel the basis for the complex effects of histone modifications on lncRNA transcription. I recommend accepting this manuscript once the following points are addressed.

We thank the reviewer for the positive remarks on our manuscript.

1. Nguyen et al. (eLife, 03635, 2014) recently reported dynamic changes in both the mRNA and lncRNA transcriptomes of budding yeast upon carbon source shifts. The authors need to compare their data with those from Nguyen et al.

The 2014 Nguyen study doesn't measure expression dynamics genome-wide (perhaps the reviewer is thinking of the Yeast Metabolic Cycle data, published by another group, to which Nguyen et al compare their results). Nguyen et al. used microarray analysis (done in the Steinmetz lab, same as our microarrays) to identify transcripts differentially expressed in glucose versus 3hr in galactose. As their genome-wide analysis does not look at Set2/Rpd3S mutants and isn't really measuring dynamics (there's only the single 3hr timepoint), a systematic comparison to our data wouldn't be very meaningful. For WT cells, we already showed that ~800 genes and many lncRNAs are induced or repressed during a time course of carbon source shifts (Kim et al., Cell, 2012). Given that the analyses were both done in the Steinmetz lab, it is very likely that the 3hr timepoint from Nguyen would agree well with our data.

However, Nguyen et al. did do a full time course of Northern blotting for the HMS2 gene cluster and showed that HMS2 sense transcription represses the overlapping antisense transcript SUT650 and a downstream gene, BAT2. Interestingly, both SUT650 and BAT2 transcripts were increased in our set2Δ analysis (see figure below). It is therefore likely that transcriptional readthrough from HMS2 targets H3K36me3 to SUT650 and BAT promoters. Compared to downstream regions of YJR149W or other intergenic regions, high levels of H3K36me3 were seen at the intergenic region between HMS2 and BAT2 (see below). The HMS2 promoter also had a high level of H3K36me3, possibly from SUT650 antisense transcription. Interestingly, HMS2 transcription was decreased in set2Δ during galactose incubation. This could be due to increased SUT650 transcription in set2Δ (double negative

effect?), as they found that insertion of a transcription terminator within the HMS2 ORF increases both HMS2 and SUT650 promoter activity.

In summary, in addition to the transcriptional interference proposed by Nguyen et al., we believe our data is consistent with targeting of Set2/Rpd3S by overlapping transcription as a contributor to negatively regulate mRNA or lncRNA transcription. We have added a few sentences in our discussion to connect our results to Nguyen et al.

2. The authors used histone modification patterns as evidence for targeting of histone modifying enzymes. This is a very indirect readout. Given that the targeting patterns of the enzymes are central to the authors' model, more direct evidence for enzyme targeting (modifying enzyme ChIP) needs to be provided. This is particularly important as the transcriptional response to mutations, which can also affect histone modifications, is highly complex.

The reviewer makes a very valid point. Unfortunately, we have been unable to get a ChIP signal for Rpd3S subunits. Nevertheless, we find that loss of Set2, loss of subunits for Rpd3S, or mutation of H3K36 to A exhibits the same phenotypes, namely increased histone acetylation at target promoters and elevated transcript levels. Albeit indirect, these results are mostly simply and logically explained by the model that H3K36 methylation by Set2 directly recruits Rpd3S activity. We note that the Hinnebusch lab suggests a slightly different model in which Rpd3S is recruited by interaction CTD, with K36 methylation only needed to allosterically activate the HDAC (Govind, Mol Cell 2010). In this model, we wouldn't expect to see a difference in Rpd3S recruitment.

3. Page9, lines 1-2: *I believe the authors' statement, "The distance between the lncRNA and mRNA promoters will determine which histone methylation predominates at the RNA promoter" is their model, rather than an established fact. However, this sentence reads as if this were a widely accepted fact. The authors need to modify the sentence.*

The sentence has been modified as “We propose that the distance between the lncRNA and mRNA promoters determines which histone methylation predominates at the mRNA promoter”

Reviewers' Comments:

Reviewer #1 (Remarks to the Author):

In this revised manuscript, Kim et al. have satisfactorily addressed my previous concerns. The manuscript describes an interesting and well-executed analysis of how mRNAs are regulated by lncRNAs through two different histone modification pathways in yeast: Set2/H3K36me3/Rpd3S and Set3/H3K4me2/Set3C. A major strength of this paper is the experimental testing of hypotheses that arose from the array studies. The inclusion of new genome-wide information on how Set2- and Set3-regulated promoters differ with respect to the distance between the lncRNA promoter and the mRNA promoter further strengthens the paper. Before publication, the authors should address the following minor concerns:

1. Figure 2C: The histone mutant data are a nice addition to the paper. However, the authors might include a sentence to explain why the WT control is showing higher AAD10 mRNA levels.
2. Line 335: The reference should be to Figure 6C
3. Figure 6C: The rows on the heat maps should be labeled.
4. Page 15: The discussion starting at line 341 is quite speculative, but some of it is written as if the authors have demonstrated the dynamic changes they discuss. For example, have the authors shown that, after two hours in glucose, H3 K36 methylation has not been re-established? I recommend that the writing be softened to clarify what has been demonstrated and what is being postulated.
5. Figure 6C legend: SC-galactose should replace SG-galactose

Reviewer #2 (Remarks to the Author):

The authors have addressed nearly all the points raised by this and other reviewers in a satisfying way, either by addition of new figures or by modifying the text. However, there is a single point that should be improved to make the paper acceptable for publication. In new Fig.5f, the authors should better describe how they performed the bioinformatic analysis, what datasets they used, and more importantly how many Set2 and Set3 repressed genes have been considered to generate the box plots.

Reviewer #3 (Remarks to the Author):

NCOMMS-16-09228A

It is unfortunate that Rpd3S ChIP did not work, as it would have made the manuscript substantially stronger. However, I agree with the authors that other evidence strongly support the roles of Rpd3S. The authors addressed other major points raised by reviewers. I believe this manuscript is now suitable for publication.

REVIEWERS' COMMENTS:

Reviewer #1 (Remarks to the Author):

In this revised manuscript, Kim et al. have satisfactorily addressed my previous concerns. The manuscript describes an interesting and well-executed analysis of how mRNAs are regulated by lncRNAs through two different histone modification pathways in yeast: Set2/H3K36me3/Rpd3S and Set3/H3K4me2/Set3C. A major strength of this paper is the experimental testing of hypotheses that arose from the array studies. The inclusion of new genome-wide information on how Set2- and Set3-regulated promoters differ with respect to the distance between the lncRNA promoter and the mRNA promoter further strengthens the paper. Before publication, the authors should address the following minor concerns:

1. Figure 2C: The histone mutant data are a nice addition to the paper. However, the authors might include a sentence to explain why the WT control is showing higher AAD10 mRNA levels.

We believe this is probably because the histone shuffling strain has replaced the two chromosomal copies of the H3 and H4 genes with a single copy on a plasmid. A sentence stating this was added on page 7.

2. Line 335: The reference should be to Figure 6C

This has been corrected.

3. Figure 6C: The rows on the heat maps should be labeled.

This has been done.

4. Page 15: The discussion starting at line 341 is quite speculative, but some of it is written as if the authors have demonstrated the dynamic changes they discuss. For example, have the authors shown that, after two hours in glucose, H3 K36 methylation has not been re-established? I recommend that the writing be softened to clarify what has been demonstrated and what is being postulated.

These sentences have been modified to make it clear this is just a proposed model.

5. Figure 6C legend: SC-galactose should replace SG-galactose

This has been corrected.

Reviewer #2 (Remarks to the Author):

The authors have addressed nearly all the points raised by this and other reviewers in a satisfying way, either by addition of new figures or by modifying the text. However, there is a single point that should be improved to make the paper acceptable for publication. In new Fig.5f, the authors should better describe how they performed the bioinformatic analysis, what datasets they used, and more importantly how many Set2 and Set3 repressed genes have been considered to generate the box plots.

This analysis has been done with the Set3 dataset (Kim et al., Cell, 2012) and the Set2 dataset provided in this study. Furthermore, 64 genes that are repressed by Set3 and 59 genes negatively regulated by Set2 were used to measure the distance between mRNA and lncRNA promoters. We have added this information in the text.

Reviewer #3 (Remarks to the Author):

It is unfortunate that Rpd3S ChIP did not work, as it would have made the manuscript substantially stronger. However, I agree with the authors that other evidence strongly support the roles of Rpd3S. The authors addressed other major points raised by reviewers. I believe this manuscript is now suitable for publication.

We thank the reviewer.